# Formulation and Evaluation of Olmesartan Medoxomil Tablets

Rocío González [ID], María Ángeles Peña *[ID] and Guillermo Torrado

Department of Biomedical Sciences, Faculty of Pharmacy, University of Alcalá (UAH), Alcalá de Henares, E-28871 Madrid, Spain
* Correspondence: angeles.pena@uah.es

**Abstract:** This work proposes a methodology for the design, development, and characterization of tablets prepared by the direct compression method of olmesartan medoxomil. The main objective was to ensure a high dissolution rate of the active ingredient. Therefore, a rigorous selection of excipients was carried out to ensure their physical and chemical compatibility with the active ingredient by scanning electron microscopy (SEM), differential scanning calorimetry (DSC) and Fourier-transform infrared spectroscopy (FT-IR) studies. The suitability of the mixture for use in direct compression was performed using SeDeM methodology. The tablets met pharmacopoeia specifications for content uniformity, breaking strength, friability, and disintegration time.

**Keywords:** olmesartan medoxomil; preformulation; tablet; physicochemical characterization

## 1. Introduction

High blood pressure, along with smoking, hypercholesterolemia and hyperglycemia, are some of the major risk factors contributing to the increase in global morbidity and mortality from cardiovascular disease [1,2]. In turn, poorly controlled hypertension is the number one risk factor for cerebrovascular disease [3]. Hypertension is defined as an increase in systolic (SBP) and diastolic (DBP) blood pressure levels considered normal (SBP > 120 mmHg and DBP > 80 mmHg) [4].

Treatment is aimed at reducing morbidity and mortality rates associated with hypertension, starting with lifestyle modifications (smoking cessation, weight reduction, reduction of salt and alcohol intake, physical exercise) in combination with antihypertensive drugs [4]. The 2018 European guidelines recommend five main classes of antihypertensive drugs: angiotensin-converting enzyme inhibitors (ACE inhibitors), angiotensin II receptor antagonists (ARA-II), β-adrenoblockers (BAB), calcium channel blockers (CCBs) and diuretics [5]. ARA-II or "sartans" potently and selectively block angiotensin II AT1 receptors resulting in vasodilation, decreased vasopressin secretion and aldosterone secretion [6,7]. ARA-IIs have demonstrated good tolerability, a high safety profile and antihypertensive efficacy with once-daily administration [8]. There are different types of ARA-IIs: olmesartan, losartan, valsartan, irbesartan and candesartan. In this research, olmesartan has been selected (5-methyl-2-oxo-1,3-dioxol-4-yl) methyl-5-(2-hydroxypropane-2-yl)-2-propyl-3-[[4-[2-[2-(2H-tetrazol-5-yl) phenyl] methyl] imidazol-4-carboxylate), a high-intensity sartan that binds with higher affinity to the AT1 receptor than the other ARA-IIs, ensuring strong and persistent blockade of angiotensin actions [9].

Olmesartan medoxomil is an inactive prodrug that after oral administration is rapidly absorbed and undergoes rapid de-esterification through the gastrointestinal tract-giving rise to the active metabolite olmesartan [10]. Due to its low water solubility, it has a low oral bioavailability (approximately 26%), thus the Biopharmaceuticals Classification System (BCS) classifies olmesartan as class II (low solubility and high permeability) [11]. Drugs administered orally in tablet form have many advantages over other dosage forms, from good stability and easy manufacture to precise dosing that facilitates adherence to treatment [12]. Therefore, the main objective of this research is the design, development, and

characterization of tablets prepared by direct compression of olmesartan medoxomil that provide a high dissolution rate of the active ingredient and thus increase its bioavailability.

The selection of tablets by the direct compression method is based on the advantages offered by this dosage form as well as the vision of subsequent industrial production as it is a simple and cost-effective manufacturing technology [13,14]. Consequently, the functionality and proportion of the excipients and the design of the production method are crucial in the compression process.

First, physical and chemical compatibility studies of olmesartan medoxomil and the selected excipients were performed by scanning electron microscopy (SEM), differential scanning calorimetry (DSC) and Fourier-transform infrared spectroscopy (FT-IR), in order to select the most suitable excipients for the formulation design [15,16].

Secondly, the SeDeM galenic methodology was applied to the pre-formulation studies of direct compression tablets to obtain information on the active ingredient-excipient mixture in terms of its suitability for use in direct compression, thus allowing for faster formulation design [17,18].

Finally, a spectrophotometric analytical method was developed to identify from the correct mixing in the production process to the concentration of olmesartan medoxomil in the tablets.

## 2. Materials and Methods

### 2.1. Materials and Reagents

Olmesartan medoxomil (Insud Pharma, Madrid, Spain) (Figure 1), lactose monohydrate (Guinama, Valencia, Spain), microcrystalline cellulose (Vivapur 12®, JRS Pharma GmbH & CO.KG, Berlin, Germany), hypromellose (HPMC 2910, JRS Pharma GmbH & CO.KG, Berlin, Germany) and magnesium stearate (Guinama, Valencia, Spain).

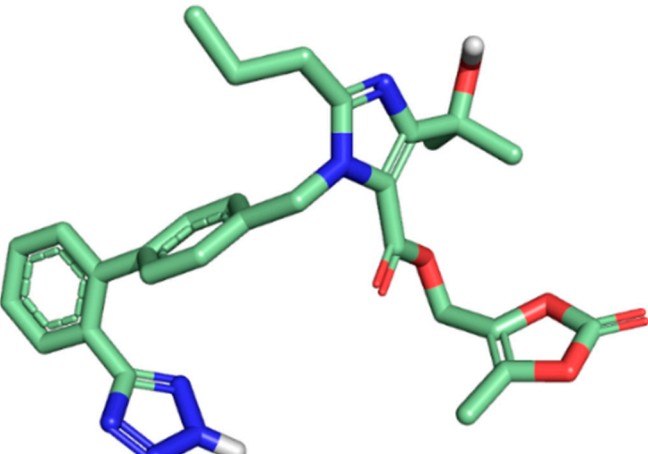

**Figure 1.** Olmesartan medoxomil 3D chemical structure. The carbon atoms are shown in green; blue, the nitrogen atoms; and the polar oxygen and hydrogen atoms in red and white, respectively.

### 2.2. Methodological Approach

The development of the definitive formulation began with a study of drugs marketed with 20 mg of olmesartan medoxomil prepared by direct compression and according to the biopharmaceutical and physicochemical characteristics of the active ingredient. There are many marketed drugs with the indicated dose, so the selection of the different excipients was made considering from their functionality to the compatibility with the active substance [19]. The search was carried out in the CIMA database (Medicine Online Information Center of Spanish Agency of Medicines and Medical Devices) and different preformulation and formulation books [20]. A matrix was made with the different declared excipients and at different proportions in order to perform compatibility and characterization studies that

allowed the formulation to be defined [21]. The suitability of the final mixture for use in direct compression was determined. Finally, the tablets were produced and characterized.

### 2.2.1. Preparation of Physical Mixtures

We weighed 1:1 proportions of olmesartan medoxomil, and each of the selected excipients, lactose monohydrate, microcrystalline cellulose, hypromellose and magnesium stearate. The components were mixed for ten minutes in a mortar ensuring homogeneity of the mixture, which was stored in a desiccator until further DSC and FT-IR analysis.

### 2.2.2. Characterization of Physical Mixtures
#### Scanning Electron Microscopy (SEM) Studies

Scanning electron microscopy (SEM) analysis was performed with the Zeiss DSM 950 (Germany) equipment using a secondary electron signal (SE) and a backscattered signal (BSE) with a resolution of 3 nm. Prior to the examination, olmesartan medoxomil and excipients were coated with gold to make them conductive of electricity.

The SEM equipment generates an electron beam of high energy that hits the material and provides a series of signals that are registered in the different detectors of the equipment, each of which provides referenced information [22–24].

#### Differential Scanning Calorimetry (DSC) Analysis

Differential scanning calorimetry analysis was performed with the Mettler TA 4000 DSC Star System equipment (Schwezenbach, Switzerland). For the analysis airtight aluminum crucibles of 40–100 μL capacity were used, the aluminum is inert with olmesartan medoxomil and the selected excipients. Samples weighing approximately 3 mg were heated at a constant rate (10 °C/min) under dynamic nitrogen gas purge (20 mL/min). Thermograms were obtained from 30 °C to 400 °C. The melting or decomposition onset temperature is used throughout the analysis, so that mass does not influence in the result [25]. Each measurement was performed in triplicate.

#### Fourier Transforms Infrared Spectroscopy (FT-IR) Analysis

The Fourier-transform infrared spectroscopy (FT-IR) analysis was performed with Fourier Spectrum 2000 spectrometer Perkin Elmer1 System 20000FT-IR (Thermo Fisher, Waltham, MA, USA). The analysis allows the quantification in different regions of the infrared spectrum of a certain type of bonds, with high sensitivity and short analysis time [26]. For the analysis, a 1:99 dilution with KBr (material to be analyzed: KBr) was homogeneously mixed in agate mortar. This mixture was taken to a press and by means of high pressure (5 T for 2 min) 13 mm diameter discs were obtained, which were subjected to study at wavelengths from 400 $cm^{-1}$ to 4000 $cm^{-1}$. The analysis was performed with olmesartan medoxomil, the selected excipients and the physical mixtures.

### 2.2.3. SeDeM Methodology

The suitability of the mixture of olmesartan medoxomil and the selected excipients for use in direct compression was determined using the SeDeM methodology by evaluating different physical properties. Five incidence factors obtained from twelve parameters (r) of the olmesartan medoxomil and excipients mixture were calculated [27].

#### Experimental Results for the SeDeM Methodology

- Apparent density (Da): the occupied volume of 10 g of the powder mixture was determined and its density in g/mL was calculated [28].
- Compressibility density (Dc): it is the volume occupied by the same amount of powder after 2500 hits on the sample. The analysis was performed in a powder density tester PT-TD200 (Pharma Test, Hainburg, Germany) and the result was expressed in g/mL [28].

- Inter-particle porosity (Ie): inter-particle porosity is calculated by means of Equation (1), dimensionless.

$$\text{Interparticle porosity (Ie)} = \frac{(\text{Dc} - \text{Da})}{\text{Dc} \times \text{Da}} \qquad (1)$$

- Carr's index (IC): IC is used to calculate the compression capacity of the powder mixture in percent (Equation (2)) [29].

$$\text{Carr's Index (IC)} = \frac{\text{Dc} - \text{Da}}{\text{Dc}} \times 100 \qquad (2)$$

- Cohesiveness's index (Icd): the hardness (resistance to breakage) was determined in a sample of five 120 mg tablets. It was used the durometer Pharmatest PTB 311 (Hamburg, Germany) and the result was reported in Newtons [30].
- Hausner's index (IH): the flow and slip capacity of the powder is calculated by means of Equation (3), which is dimensionless [29].

$$\text{Hausner's Index (IH)} = \frac{\text{Dc}}{\text{Da}} \qquad (3)$$

- Angle of repose ($\alpha$): a funnel 9.5 cm high, 7.2 cm in diameter of the upper mouth and 1.8 cm in diameter of the lower mouth is placed in a support at 20 cm from the surface of the test. The lower mouth of the funnel is covered, and it is filled with the powder mixture. Then the lower mouth is uncovered to allow the powder to exit the funnel. The height of the cone ($h$), the four radiuses of the base of the cone formed were measured and the average value of the radiuses ($r$) were calculated. The angle was determined by Equation (4) [29,31].

$$\text{tg}(\alpha) = \frac{h}{r} \qquad (4)$$

- Sliding time (t''): the time it takes to pass 10 g of powder mixture through a funnel to the surface is timed. If the powder does not flow, it is rated seconds [32].
- Relative humidity (%HR): humidity was determined by calculating the difference in weight of a 4 g sample of powder before and after oven drying. The analysis was carried out at 105.0 °C ± 2.0 °C during 2 h using the Rayna Liebherr FKS1800 oven type 200041 (Bad Schussenried, Germany). The different in percent is the %HR [33].
- Hygroscopicity (%H): it determines the weight increase of the sample after being kept in a humidifier at 76.0% ± 2.0% relative humidity and 22.0 °C ± 2.0 °C temperature for 24 h, the different in percent is the %H [33].
- Determination of percentage of particles <50 μm (%Pf): it was calculated the % of powder particles passing through a 50 μm mesh size sieve while vibrating for 10 min at level three on a vibrating shaker for cascade of CISA sieves (Biotech, Barcelona, Spain). A 20 g sample of dust was weighed [34].
- Homogeneity index (Iθ): a sample of 50 g of powder mixture was subjected to a sieve scale with a 10 min vibration at level three. The sieves used are 355 μm, 212 μm, 100 μm, 50 μm of light placed in increasing order (Equation (5)) [34,35].

$$I\theta = \frac{Fm}{100 + (dm - dm - 1) \times Fm - 1 + (dm + 1 - dm) \times Fm + 1 + (dm - dm - n) \times Fm - n + dm + n - dm) \times Fm + n} \qquad (5)$$

Fm: percentage of particles in the majority range,
Fm − 1: percentage of particles in the range immediately below the majority range,
Fm + 1: percentage of particles in the range immediately above the majority range,
n: order number of the fraction studied under a series, with respect to the majority fraction,
dm: mean diameter of the particles in the majority fraction,
dm − 1: mean diameter of the particles in the fraction of the range immediately below the majority range,

dm + 1: mean diameter of the particles in the fraction of the range immediately above the majority range.

From the 12 experimental results, the parameters (r) were calculated using a conversion factor that expressed the "r values" obtained on a scale from 0 to 10 (Table 1). The "r values" influence the incidence factors that determine the suitability for use in direct compression, these values were graphically represented in the SeDeM diagram (Figure 2) [17,36].

**Table 1.** Conversion factors to obtain "r values".

| Parameter | Conversion Factor | Parameter (r) |
|---|---|---|
| Apparent density (Da) | $10 \times Da$ | rDa |
| Compressibility density (Dc) | $10 \times Dc$ | rDc |
| Interparticle porosity (Ie) | $(10 \times Ie)/1.2$ | rIe |
| Carr's index (IC) | $IC/5$ | rIC |
| Cohesiveness's index (Icd) | $Icd/20$ | rIcd |
| Hausner's index (IH) | $5 \times (3 - IH)$ | rIH |
| Angle of repose ($\alpha$) | $10 - (\alpha/5)$ | r$\alpha$ |
| Slidding time (t'') | $10 - (t''/2)$ | rt'' |
| Relative humidity (%HR) | $10 - \%HR$ | r%HR |
| Higroscopicity (%H) | $10 - (\%H/2)$ | r%H |
| Determination of percentage of particles <50 μm (%Pf) | $10 - (\%Pf/5)$ | r%pf |
| Homogeneity index (Iθ) | $500 \times I\theta$ | rIθ |

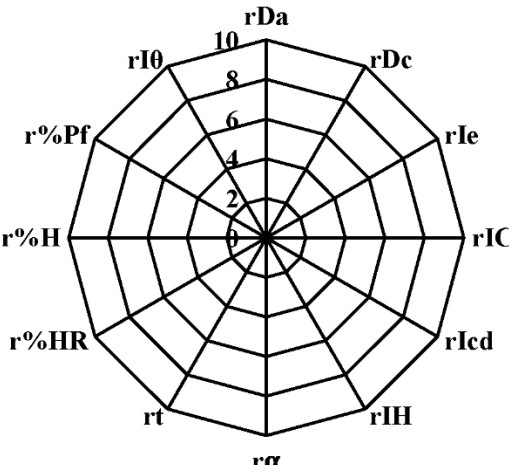

**Figure 2.** SeDeM diagram for the calculated parameters (r).

Incidence Factors for the SeDeM Methodology

- Dimensional impact factor ($F_{dimens}$): ability of the powder mixture to compact and the consequences on tablet dimensions (Equation (6)).

$$F_{dimens} = \text{Average (rDa; rDc)} \tag{6}$$

- Compressibility impact factor ($F_{compressib}$): ability of the powder mixture to be compacted and maintain its shape (Equation (7)).

$$F_{compressib} = \text{Average (rIe; rIC; rIcd)} \tag{7}$$

- Incidence factor of slippage/fluidity ($F_{flowability}$): flowability of the powder mixture (Equation (8)).

$$F_{flowability} = \text{Average (rIH; r}\alpha\text{; rt)} \tag{8}$$

- Incidence factor of lubricity/stability ($F_{lub/stability}$): consequence of residual moisture and hygroscopicity of the powder mixture on sliding and compaction (Equation (9)).

$$F_{lub/stability} = Average\ (r\%HR; r\%H) \tag{9}$$

- Incidence factor of lubricity/dosage ($F_{lub/dosage}$): Consequence of the powder particle size distribution on the sliding and correct filling of the compression matrices (Equation (10)).

$$F_{lub/dosage} = Average\ (r\%Pf; rI\theta) \tag{10}$$

Incidence Factors to Determine the Capacity to Be Used in Direct Compression for the Sedem Methodology

Finally, three parameters were calculated that determine the ability of the powder mixture to be used in direct compression [17,18,36]:

- Parametric index (IP) (Equation (11)).

$$IP = \frac{n° \ P \geq 5}{n° \ Pt} \tag{11}$$

n° P ≥ 5: n° of parameters (r) whose value is ≥ 5,
n° Pt: n° total parameters (r) studied.
The minimum expected value for a good powder mixture is IP ≤ 0.5.

- Parametric profile index (IPP) (Equation (12)).

$$IPP = \frac{\sum_{i=1}^{n} ri}{n} \tag{12}$$

$\sum_{i=1}^{n} ri$ : sum of all parameters (r) studied,
n: n° total parameters (r) studied.
The minimum expected value for a powder with suitable characteristics is IPP ≤ 5.

- Good compression index (IGC) (Equation (13)).

$$IGC = IPP \times f \tag{13}$$

IPP = parametric profile index,
f = reliability factor (f = 0.952).
The expected value for considering a powder suitable for direct compression is IGC ≥ 5.

### 2.2.4. Preparation and Characterization of Tablets

### Preparation of Tablets Formulation

A laboratory batch of 400 tablets weighing 120 mg was manufactured. For this purpose, a CISA sieve shaker (Barcelona, Spain) was used for the sieving process, a V-blender (Biotech, Barcelona, Spain) for the mixing process, and a Pharma Test durometer (PTB 311, Pharma Test, Hamburg, Germany) was used to test the hardness of the tablet during the compression process. Tablets were manufactured using an eccentric compression machine J. Bonals1 40B type MT (Bonals technologies, Barcelona, Spain). Compression was unidirectional using non-grooved flat-faced punches and die 6 mm in diameter.

The mixing and compression stages are crucial to ensure quality, safe and effective drugs, therefore, the selection of critical parameters allows to guarantee that the manufacturing process is ideal, consequently, Quality by Design (QbD) was applied in this research. The QbD approach enabled the identification of the quality target product profile (QTPP), which includes critical quality attributes (CQA), identifying and optimizing critical material attributes (CMA), as well as identifying critical process parameters (CPP) that affect product performance [37,38] (Figure 3).

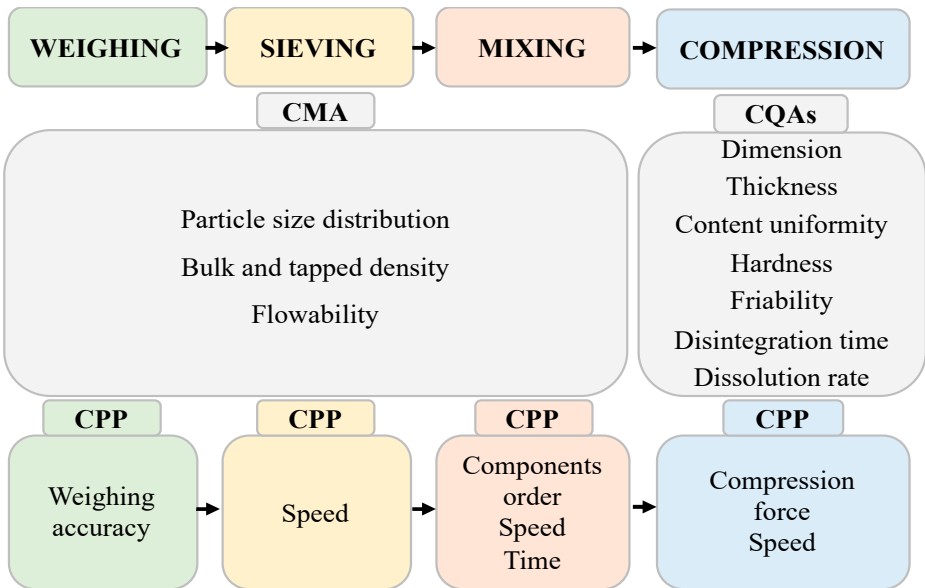

**Figure 3.** Process of the study.

Tablets Characterization

- Diameter and Thickness

The diameter and thickness were measured using Tablet Testing Instrument Pharmatest PTB 311 (Pharma Test, Hamburg, Germany). It was performed using a ten-tablets sample.

- Content Uniformity

Ten tablets were accurately weighed individually using an analytical balance Mettler Toledo AG 245 (Schwerzenbach, Switzerland). Using an appropriate analytical test, the percentage of drug in each tablet was determined and the acceptance value (AV) was calculated using Equation (14). According to European Pharmacopoeia specifications [39], AV should be less than L1 = 15.

$$AV = |M - X| + ks \tag{14}$$

X: Mean of individual contents expressed as a percentage of the label claim.
M: Reference value, in this case, 99.9.
k: Acceptability constant. In this case, 2.4.
s: Sample standard deviation.

- Hardness

The hardness or resistance to breakage was determined in 10 tablets individually. It is defined as the force in Newton required causing them to break by crushing. We used the durometer Pharmatest PTB 311 (Hamburg, Germany).

- Friability Test

Friability was determined on a sample of 20 tablets using the Pharmatest PTF E1 friabilometer (Pharma Test, Hamburg, Germany). The weight of the 20 tablets (Pinitial) was calculated, and after rotating 100 times at 25 rpm in the friabilometer, the weight of the 20 tablets (Pfinal) was determined again [40]. From the Pinitial and Pfinal weight, the percentage weight loss was calculated (Equation (15)).

$$\%\text{friability} = \frac{\text{Pinitial} - \text{Pfinal}}{\text{Pinitial}} \times 100 \tag{15}$$

- Disintegration Time

The disintegration test determined the time taken for the tablets to disintegrate completely using the Turu-Grau disintegration equipment (Spain) with distilled water at 37.0 ± 0.5 °C. The test was carried out with a sample size of six tablets [41].

- Dissolution Rate Study

The dissolution rate study was performed to determine the dissolution kinetics of olmesartan tablets. Quantitative analysis of olmesartan medoxomil in the formulation tablets was performed by a standard line using infrared absorption spectrophotometry. Standards were prepared from a stock solution of 0.07 mg olmesartan/mL using a buffer solution at pH = 6.8. The standards were analyzed at a wavelength of 250 nm. The validation of the spectrophotometric method was performed by evaluating the following parameters: linearity, accuracy, sensitivity, and precision [42,43].

(1) Linearity: linearity was determined using the least squares method, the absorbance of the standards was related to their concentration. Linearity is acceptable with a coefficient of determination ($r^2$) greater than 0.995.

(2) Accuracy: accuracy (expressed as recovery, %) was assessed by relating the theoretical concentration (A) to the real concentration (B) using the following formula: $A/B \times 100\%$. Each sample was evaluated in triplicate.

(3) Sensitivity: sensitivity was determined by calculating the limits of detection (LOD) and quantification (LOQ). The LOD is the lowest concentration of analyte that can be detected with precision and accuracy. The LOQ is the lowest concentration of analyte that can be quantified with precision and accuracy. Thus, it was evaluated the minimum amount of analyte needed to obtain a meaningful result.

(4) Precision: precision was evaluated in triplicate at four concentration levels (8.40; 11.20; 14.00 and 16.00 µg/mL). To determine the precision of the analytical method the coefficient of variation must be less than 2%.

The dissolution rate test of the tablets was performed with the Hanson Research SR8 SRII 8-Flak (SpectraLab Scientific, Markham, Canada). First, 900 mL of 0.2 M phosphate buffer pH 6.8 was used in a water bath at $37.0 \pm 0.5\,^{\circ}\text{C}$ as dissolution medium. The rotation speed of the paddles was $50 \pm 2$ rpm. The samples were passed through an inert filter of adequate porosity (0.45 µm) and did not retain, to any significant degree, the dissolved active principle [44]. Finally, the dissolved drug was analyzed by UV-Vis spectrophotometry at a wavelength of 250 nm (Spectronic Helios Gamma UV-Vis spectrophotometer, (Thermo Fisher, Waltham, MA, USA). Dissolution analysis was performed on six tablets individually.

## 3. Results and Discussion

The excipients selected for the development of olmesartan medoxomil tablets were diluents, binding agents, disintegrants and lubricants. Therefore, the final composition of olmesartan tablets was olmesartan medoxomil (17%), lactose monohydrate (66%), microcrystalline cellulose (9%), hypromellose (5%) and magnesium stearate (3%) (Table 2).

**Table 2.** Composition of tablet formulation expressed in mg per tablet.

| Olmesartan Medoxomil | Drug | 20.0 mg |
|---|---|---|
| Lactose monohydrate | Diluent | 79.0 mg |
| Microcrystalline cellulose (Vivapur 12®) | Diluent, binding agent, disintegrant | 11.0 mg |
| Hypromellose 2910 (Hydroxypropylmetylcellulose, HPMC) | Binding agent | 6.0 mg |
| Magnesium stearate | Lubricant | 4.0 mg |
| TOTAL TABLET | | 120.0 mg |

Lactose monohydrate provided excellent flow properties to the formulation, however, it has moderate compactness [21,45], so it was used in combination with microcrystalline cellulose for its good binding properties [46]. Microcrystalline cellulose is composed of porous particles of different sizes and moisture content, which give it different properties and applications [36]. Vivapur 12® was chosen for this formulation. Due to its particle size (180 µm) it has good flow properties [46]. Hypromellose is a partially O-methylated and O-(2-hydroxypropyled) cellulose ether. Depending on the proportions of methoxy groups, hydroxypropyl groups and molecular weight, HPMC grades with characteristics, behavior, and properties different are classified [47]. The HPMC used corresponds to hypromellose 2910, which, due to the percentage of methoxy groups, has a high viscosity. Magnesium stearate was used for its good non-stick properties, preventing the formulation from sticking to the punches and the die of the compression machine, as well as for its excellent lubricating action that

reduced friction between the particles during the compression process. Its effectiveness is achieved at concentrations of 0.25–5.0% *w/w* [46,48].

The mixing process started with mixing lactose monohydrate and Vivapur 12® for five minutes at 30 rpm conditions. After that, hypromellose 2910 and olmesartan medoxomil were added for five minutes at 30 rpm finally magnesium stearate was added for three minutes at 30 rpm. Compression was performed at eight pressures on the bonal scale with a 6 mm punch. The pharmaceutical technological characteristics were determined according to the indications of the Royal Spanish Pharmacopoeia (R.F.E.).

### 3.1. Solid-State Characterization

SEM, DSC, and FT-IR determined the possible changes and interactions between olmesartan medoxomil and the selected excipients The combination of these techniques provides information about possible incompatibilities between the formulation components. The use of this technique ensured quality, safety, and protection [15].

### 3.1.1. SEM Studies

Scanning Electron Microscopy (SEM) provided topographical, structural conductivity, and compositional information on olmesartan medoxomil and the excipients (Figure 4). Due to the high resolution, it was known from the crystalline structure to size distribution, porosity, and surface morphology of the components.

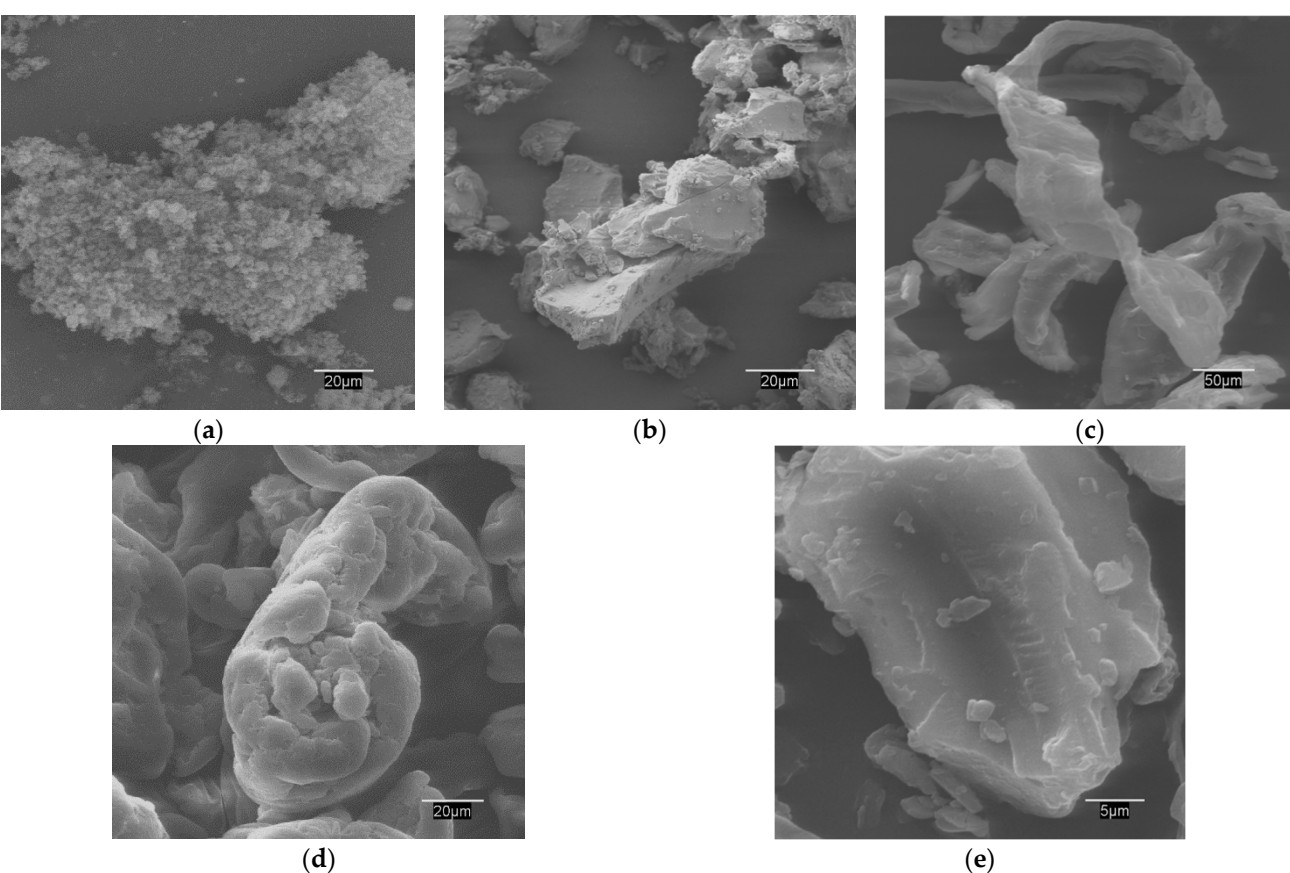

**Figure 4.** SEM of olmesartan medoxomil and selected excipients. (**a**) olmesartan medoxomil 600×; (**b**) lactose monohydrate 700×; (**c**) Vivapur 12® 500×; (**d**) hypromellose 2910 625×; (**e**) magnesium stearate 2400×.

Olmesartan medoxomil is a hydrophobic drug with an irregular morphology and a crystalline appearance (Equation (2)) [49]. Lactose monohydrate is found as granulated/agglomerated with small amounts of anhydrous lactose, thus allowing for its possible use in low doses of drug without granulation (Equation (3)) [46]. Vivapur® 12 is a thick grade of microcrystalline cellulose (180 μm) with high binding capacity, good compactibility, and excellent flow properties (Equation (4)) [50].

Hypromellose 2910 has a smooth, homogeneous surface and a rounded shape, favoring dispersion and release of the drug. Finally, magnesium stearate is a very fine, white powder with irregular edges.

### 3.1.2. DSC Studies

Differential scanning calorimetry (DSC) determined the amount of heat absorbed or released by olmesartan medoxomil and excipients when they subjected to a constant temperature for a given time, resulting in an endothermic or exothermic process. First, the individual behavior of the active pharmaceutical ingredient and excipients was determined (Figure 5). Differential enthalpic analysis of olmesartan medoxomil exhibited a single endothermic peak located at $T_{onset} = 181.54\ ^{\circ}C$ ($\Delta F = 91.97\ J/g$), corresponding to its melting, which is characteristic of its crystalline nature.

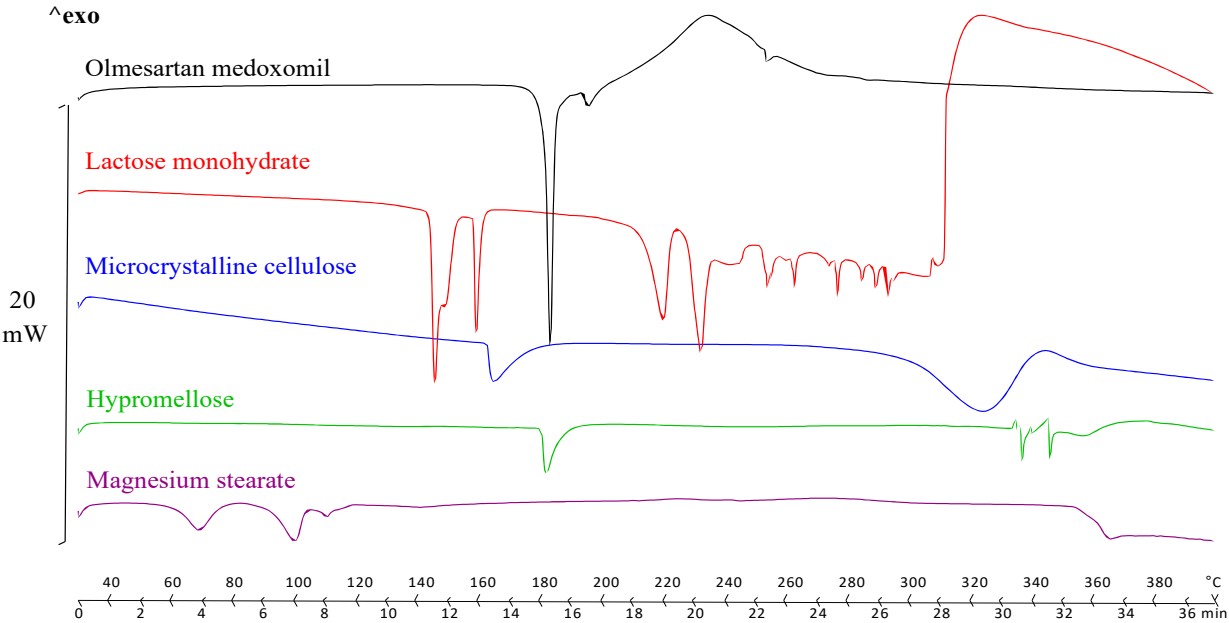

**Figure 5.** Differential scanning calorimetry (DSC) olmesartan medoxomil and excipients lactose monohydrate (red), microcrystalline cellulose (blue), hypromellose (green), and magnesium stearate (purple).

In the DSC study of α-lactose monohydrate, several endothermic peaks were observed. The first two are a consequence of water loss: the first ($T_{onset} = 144.44\ ^{\circ}C$) corresponds to the loss of surface water and the second ($T_{onset} = 158.40\ ^{\circ}C$) is associated with the loss of water of crystallization [51], according to the literature, α-lactose monohydrate releases its water of crystallization above 150 °C. The third peak ($T_{onset} = 214.45\ ^{\circ}C$) is due to the melting of α-lactose monohydrate crystals. The last peak ($T_{onset} = 228.38\ ^{\circ}C$) is the result of the presence of small amounts of β-lactose anhydrous, the thermal event is due to thermal degradation of the polymorph (Figure 5). In the thermal analysis of microcrystalline cellulose, a first endothermic peak ($T_{onset} = 162.86\ ^{\circ}C$) was observed, which is attributed to water evaporation (acid hydrolysis of crystalline cellulose) (Equation (5)) [52]. However, crystallization is achieved by increasing the hydrolysis time, continuing the heating up to 301.46 °C (Figure 5). Hypromellose is an amorphous polymer with a single endothermic peak corresponding to the glass transition temperature ($T_g$) at 180.70 °C (Figure 5) (Equation (6)) [53]. Finally, magnesium stearate showed two first peaks corresponding to water evaporation at 61.72 °C and 93.02 °C, followed by a third peak at $T_{onset} = 107.97\ ^{\circ}C$ which is due to the melting of magnesium palmitate, since stearic acid and palmitic acid are present in its composition.

Figure 6 describes the results of binary mixtures (1:1) of olmesartan medoxomil and each of the selected excipients, to determine possible interactions and incompatibilities between them through the appearance, disappearance and/or displacement of peaks, as well as variations in enthalpy values (Equation (7)) [54].

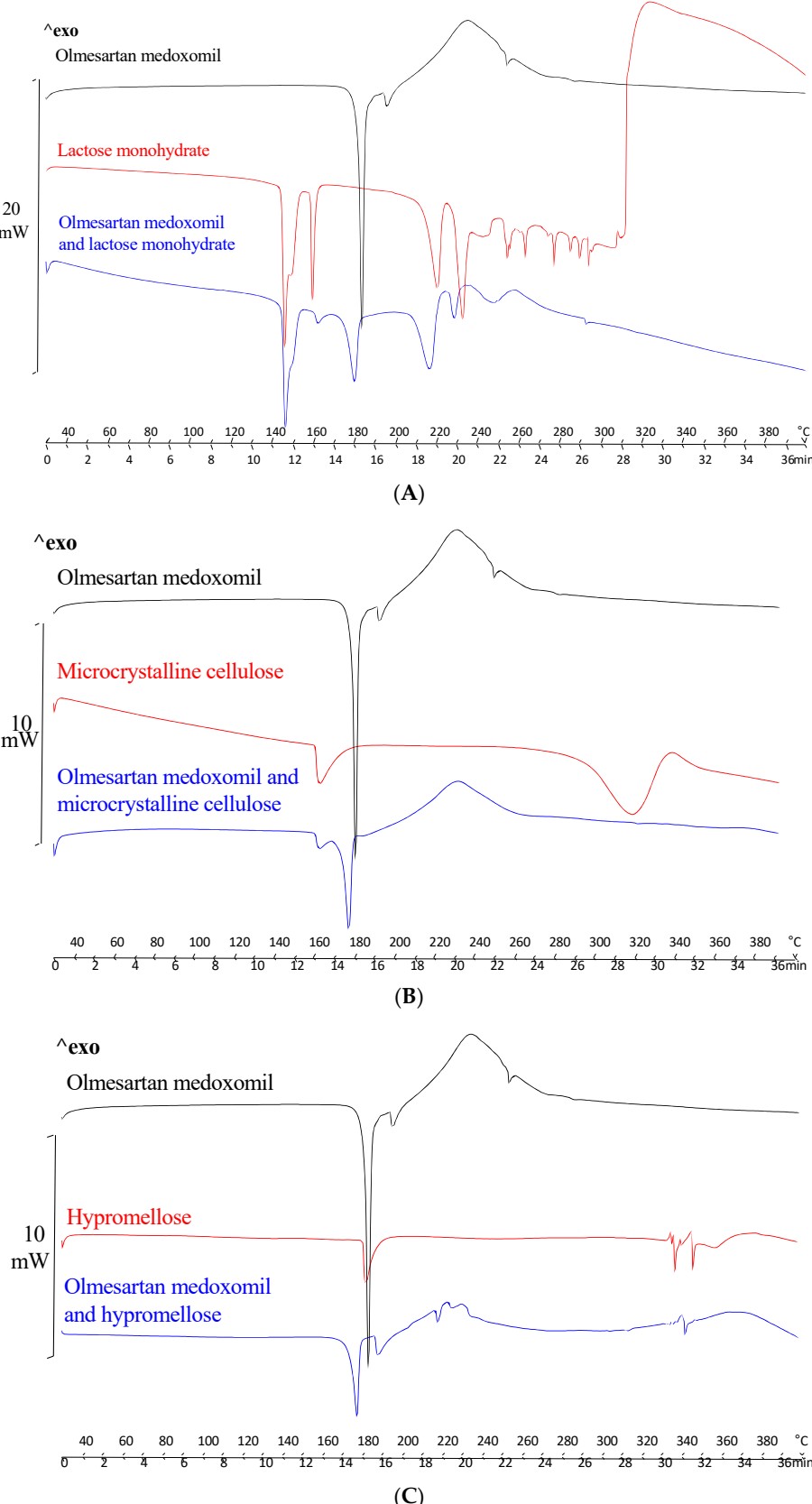

**Figure 6.** *Cont.*

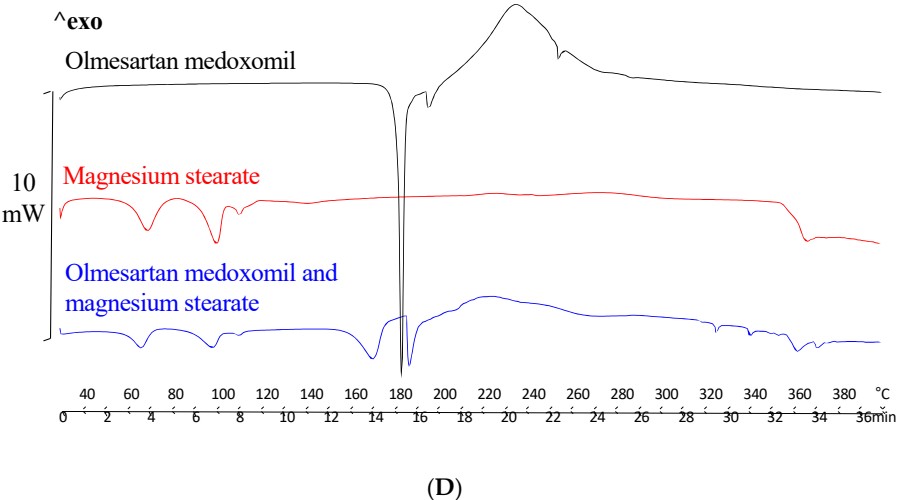

**(D)**

**Figure 6.** (**A**) Differential scanning calorimetry (DSC) of olmesartan medoxomil, lactose monohydrate and physical mixture; (**B**) DSC of olmesartan medoxomil, microcrystalline cellulose and physical mixture; (**C**) DSC of olmesartan medoxomil, hypromellose and physical mixture; (**D**) DSC of olmesartan medoxomil, magnesium stearate and physical mixture.

Figure 6A,B show the results corresponding to the physical mixture with lactose monohydrate and microcrystalline cellulose (Vivapur 12®), respectively. In both cases the endothermic peaks appearing in the DSC curves of pure olmesartan are reproduced. The small variations in the enthalpy of fusion of the drug are due to the mixing of the drug with the excipients, resulting in a decrease in the purity of each component, but in neither case do they indicate a potential incompatibility. Despite the presence of amines in the structure of olmesartan, the Maillard reaction does not occur when mixed with lactose monohydrate, as the Maillard reaction occurs on an amine basis and not in the presence of a salt. Figure 6C shows the results for the physical mixture with hypromellose, both melt at almost the same temperature 181.54 °C (olmesartan medoxomil) and 180.74 °C (hypromellose), with a masking of the melting point of both in the physical mixture. In Figure 6D, physical mixture 1:1 olmesartan: magnesium stearate, there was a decrease in the melting of olmesartan at 162.37 °C ($\Delta F = 23.16$ J/g), as well as changes in the shape of the peak, which may suggest the presence of an interaction between the two as observed in other studies (Equation (8)) [55]. Magnesium stearate forms a surface film around the olmesartan particles (Equation (9)) [48], this binding between active substance and excipient may facilitate the lowering of the melting point of olmesartan.

### 3.1.3. FT-IR Studies

The infrared spectrum of olmesartan medoxomil (Figure 7) revealed characteristic absorption peaks as previously published ensuring the presence of certain functional groups (Equation (10)) [56]. A characteristic peak at 3293.04 cm$^{-1}$ due to N–H stretching vibrations, two sharp peaks at 1832.92 cm$^{-1}$ and 1708.64 cm$^{-1}$ characteristic of the carbonyl group (C=O) were observed. The signals 1553.19 cm$^{-1}$, 1532.53 cm$^{-1}$ and 1474.86 cm$^{-1}$ are due to C=C stretching of the aromatic. In addition, the spectrum showed six peaks for the C–O strain, 1302.87 cm$^{-1}$, 1227.04 cm$^{-1}$, 1169.36 cm$^{-1}$, 1136.44 cm$^{-1}$, 1089.6 cm$^{-1}$, 1054.22 cm$^{-1}$.

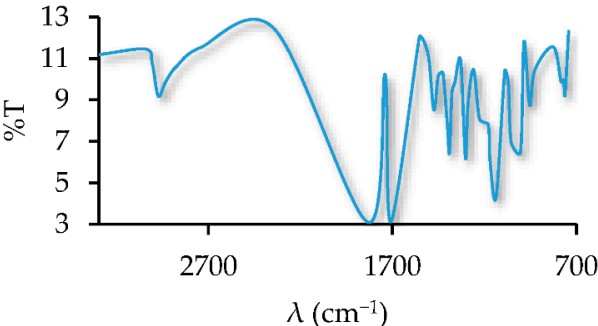

**Figure 7.** IR spectrum of olmesartan medoxomil.

A comparison was made between the physical mixtures (1:1) active substance: excipient and the infrared spectra of the pure raw materials, showing that the mixtures with lactose monohydrate (Figure 8A) and hypromellose (Figure 8B) showed a greater variation.

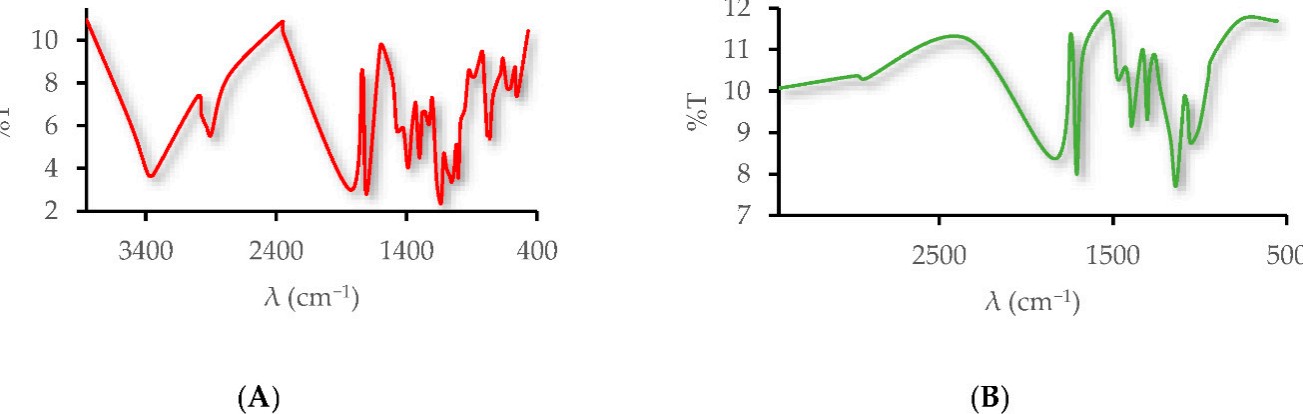

**(A)** **(B)**

**Figure 8.** (**A**) IR spectrum of physical mixture olmesartan medoxomil and lactose monohydrate; (**B**) IR spectrum of physical mixture olmesartan medoxomil and hypromellose.

Table 3 shows the values of the peaks closest to the area where the greatest variation is observed. The peak 3421.32 cm$^{-1}$ of the physical mixture with hypromellose and the peaks 3382.54 cm$^{-1}$ and 3342.96 cm$^{-1}$ of the physical mixture with lactose monohydrate are higher than 3293.04 cm$^{-1}$ of pure olmesartan medoxomil. The reason could be the formation of hydrogen bonds (Equation (11)) [57].

**Table 3.** Peaks (cm$^{-1}$) of olmesartan, physical mixture: hypromellose and physical mixture: lactose monohydrate.

| Olmesartan (cm$^{-1}$) | Olmesartan: Hypromellose (cm$^{-1}$) | Olmesartan: Lactose Monohydrate (cm$^{-1}$) |
|---|---|---|
| - | - | 3382.54 |
| - | 3421.32 | 3342.96 |
| 3293.04 | - | - |
| 3041.22 | - | - |
| 3005.49 | - | 3006.05 |
| 2973.52 | 2972.71 | 2975.95 |

### 3.2. Characterization of Powder Blends

The suitability of the mixture of olmesartan medoxomil and the excipients selected for use in direct compression was determined using the SeDeM plot. The pharmacotechnical parameters determined experimentally were: dimensional parameter, bulk density (rDa) and compacted density (rDc); compressibility parameter, interparticle porosity (rIe), Carr's index (rIC) and Cohesion index (rIcd); flow parameter, Hausner index (rIH), angle of repose (rα) and sliding time (rt); lubricity/stability parameter, relative humidity (r%RH) and hygroscopicity (r%H); and, finally, lubricity/dosage parameter, particle size < 50 μm (r%Pf) and homogeneity index (rIθ) (Table 4). The results were mathematically processed for subsequent graphical representation in the form of a SeDeM diagram (Figure 9).

According to the results obtained from the different parameters and tests that constitute the SeDeM system, the mixture of olmesartan medoxomil, lactose monohydrate, microcrystalline cellulose, hypromellose and magnesium stearate was considered suitable for use in direct compression formulations.

The acceptance indexes gave results > 5 for the parametric profile index (5.37) and good compression index (5.12), as well as results above 0.5 for the parametric index (0.58) indicating that the blend is suitable for direct compression. In turn, the values of bulk density and compacted density were close to 0.5 g/mL (Da = 0.45 g/mL) and above this value (Dc = 0.65 g/mL).

**Table 4.** Parameters and tests used by the SeDeM method.

| Experimental Results | | | | | | | | | | | |
|---|---|---|---|---|---|---|---|---|---|---|---|
| Da (g/mL) | Dc (g/mL) | Ie | IC (%) | Icd (N) | IH | α (°) | t'' | %HR | %H | %Pf | Iθ |
| 0.45 | 0.65 | 0.66 | 30 | 49.03 | 1.43 | 42.46° | ∞ | 1.32 | 0.92 | 5.21 | 0.006 |
| Parameters (r) | | | | | | | | | | | |
| rDa | rDc | rIe | rIC | rIcd | rIH | rα | rt | r%HR | r%H | r%Pf | rIθ |
| 4.55 | 6.50 | 5.50 | 6.00 | 2.45 | 7.85 | 1.51 | 0.00 | 8.68 | 9.54 | 8.96 | 2.95 |
| Impact Factor | | | | | | | | | | | |
| Dimensional | | Compressibility | | | Flowability | | | Lubricity/Stability | | Lubricity/Dosage | |
| 5.53 | | 4.65 | | | 3.12 | | | 9.11 | | 5.95 | |
| Index | | | | | | | | | | | |
| IP | | | | IPP | | | | IGC | | | |
| 0.58 | | | | 5.37 | | | | 5.12 | | | |

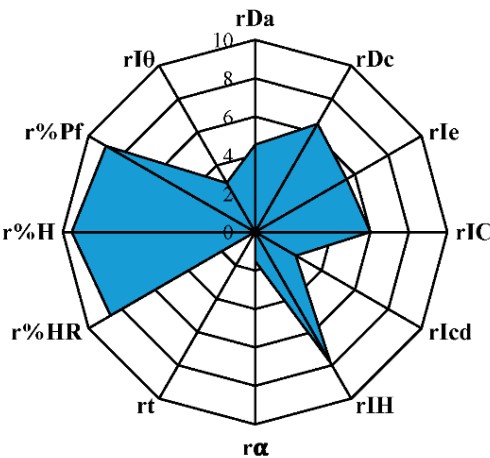

**Figure 9.** SeDeM olmesartan for the calculated parameters (r).

It is of great importance to highlight the results involved in the lubricity/stability incidence factor with values above 5. The mixture showed from excellent rheological qualities to good stability because of the low percentage of hygroscopicity (0.92%) and relative humidity (1.32%). Thus, the mixture provided excellent flow and compression set, thanks also to the low percentage of particles below 50 μm (8.96%).

### 3.3. Critical Quality Attributes of Olmesartan Medoxomil Tablets

Table 5 describes the critical quality attributes (CQA) of olmesartan medoxomil 20 mg tablets obtained by direct compression. It guarantees that the manufacturing process is suitable and ensures the desired quality of the product with compliance to pharmacopoeia stabilized specifications. Physical characteristics, dimensions, content uniformity, breaking strength, friability, disintegration time and dissolution rate were determined.

The olmesartan medoxomil 20 mg tablets obtained had a bright white visual appearance. Due to the 6.0 mm punch used, they had a diameter of 6.0 mm, a thin thickness of 3.5 mm, and were non-grooved.

The tablets were found to comply with pharmacopoeia content uniformity specifications with an acceptance value of less than 15 (4.56). Similarly, RFE 2.9.7 (Equation (12)) [40] states that uncoated tablets should have a percentage weight loss of less than 1%, olmesartan tablets meet the specifications with a deviation of 0.27%. On the other hand, the force applied during the compression process was ideal; the tablets obtained had a breaking strength of 49.03 Nw. Finally, the disintegration of the tablets should be highlighted with a disintegration profile of less than 20 s.

**Table 5.** Critical quality attributes of olmesartan medoxomil tablets.

| Critical Quality Attributes | Olmesartan Medoxomil Tablets 20 mg |
|---|---|
| Physicals characteristics, dimension, thickness | Bright White<br>$\varnothing$ = 6.00 mm<br>T = 3.50 mm<br>(1) |
| Content uniformity | AV = 4.56<br>(2) |
| Hardness | 49.03 N |
| Friability | $W_0$ = 2.58 g<br>$W_f$ = 2.57 g<br>D = 0.27%<br>(3) |
| Disintegration Time | Between 10.00–15.00 s |
| Dissolution Rate | Nearly 100% |

(1) Diameter ($\varnothing$) medium of 10 units. Thickness (T) medium of 10 units. (2) Acceptance value (AV). (3) W0 = initial weight; Wf = final weight; D = deviation.

### 3.3.1. Dissolution Rate Study

- Quantification of Olmesartan Medoxomil

Quantitative analysis of olmesartan medoxomil was determined by least squares linear regression (Equation (13)) [58]. The concentration range analyzed was from 2.80 µg/mL to 22.40 µg/mL for the active substance olmesartan medoxomil (Table 6).

**Table 6.** Parameters obtained in the production of the standard straight line.

| Linear Range (µg/mL) | Slope ($\frac{absorbance}{µg/mL}$) | Intercept (Absorbance) | Coefficient of Determination ($r^2$) |
|---|---|---|---|
| 2.80–22.40 | 0.043 | 0.0045 | 0.997 |

### 3.3.2. Method Validation

The linearity, accuracy, sensitivity, and precision results are detailed below.

- Linearity.

The analytical method was linear for the range of concentrations analyzed. With a linear coefficient of determination ($r^2$) of 0.997 (Table 6) it conforms to the acceptance criteria ($r^2 \geq 0.995$).

- Accuracy and Sensitivity.

The method was accurate over the range of concentrations studied with values within the permitted acceptance limits (85–115%). Accuracy was expressed as recovery (%). Each standard was evaluated in triplicate and the mean results ranged from 97% to 106% (Table 7). Similarly, the analytical method demonstrated high sensitivity with very low limits of detection (LOD = 0.64 µg/mL) and quantification (LOQ = 2.14 µg/mL). The results demonstrate that, using this analytical method, olmesartan can be detected and quantified even at very low concentrations (Table 7).

- Precision.

The precision of the method was expressed as coefficient of variation (CV), with values within the permitted acceptance limits, CV $\leq$ 2%. It was analyzed at four concentration levels: 8.40; 11.20; 14.00 and 16.00 µg/mL. The method was precise with a total coefficient of variation of 1% (Table 8).

### 3.3.3. Dissolution Profile of Tablets

The dissolution profile of the olmesartan tablets is shown in Table 9. A gradual release of the active substance was observed, as well as a high dissolution rate with more than 90% of the drug dissolved after 40 min.

**Table 7.** Parameter obtained in the determination of the accuracy, sensitivity, LOD and LOQ of the analytical method.

| Theorical Concentration (µg/mL) | Average Abs. | Real Concentration (µg/mL) | Accuracy [a] | Sensitivity |
|---|---|---|---|---|
| 2.80 | 0.128 | 2.86 | 102 | 0.046 |
| 5.60 | 0.246 | 5.58 | 100 | 0.044 |
| 7.00 | 0.302 | 6.88 | 98 | 0.043 |
| 8.40 | 0.359 | 8.19 | 97 | 0.043 |
| 11.20 | 0.486 | 11.13 | 99 | 0.043 |
| 14.00 | 0.600 | 13.75 | 98 | 0.043 |
| 16.00 | 0.736 | 16.90 | 106 | 0.046 |
| 19.00 | 0.841 | 19.33 | 102 | 0.044 |
| 21.00 | 0.899 | 20.65 | 98 | 0.043 |
| 22.40 | 0.959 | 22.05 | 98 | 0.043 |
| %SD | | | | 0.001 |
| LOD (µg/mL) | | | | 0.64 |
| LOQ (µg/mL) | | | | 2.14 |

[a] Expressed as recovery, %.

**Table 8.** Evaluation of the precision of the analytical method.

| Concentration (µg/mL) | Absorbances | | | Average Abs. | % RSD | CV (%) |
|---|---|---|---|---|---|---|
| 8.40 | 0.364 | 0.355 | 0.358 | 0.359 | 0.005 | 1 |
| 11.20 | 0.491 | 0.492 | 0.476 | 0.486 | 0.009 | 2 |
| 14.00 | 0.607 | 0.596 | 0.597 | 0.600 | 0.006 | 1 |
| 16.00 | 0.743 | 0.745 | 0.721 | 0.736 | 0.013 | 2 |
| | | | | | CV average | 1 |

**Table 9.** Dissolution profile of olmesartan tablets.

| Time (min) | Absorbances | | | | | | Average Abs. | Concentration (µg/mL) | Standard Deviation | CV (%) | % Dissolved |
|---|---|---|---|---|---|---|---|---|---|---|---|
| | Tablet 1 | Tablet 2 | Tablet 3 | Tablet 4 | Tablet 5 | Tablet 6 | | | | | |
| 0 | 0.000 | 0.000 | 0.000 | 0.000 | 0.000 | 0.000 | 0.000 | 0.00 | 0.000 | 0 | 0% |
| 5 | 0.248 | 0.205 | 0.177 | 0.267 | 0.265 | 0.293 | 0.243 | 5.52 | 0.043 | 18 | 25% |
| 10 | 0.489 | 0.384 | 0.343 | 0.458 | 0.501 | 0.504 | 0.447 | 10.22 | 0.068 | 15 | 46% |
| 15 | 0.629 | 0.556 | 0.497 | 0.602 | 0.646 | 0.637 | 0.595 | 13.63 | 0.058 | 10 | 62% |
| 20 | 0.695 | 0.650 | 0.590 | 0.673 | 0.703 | 0.729 | 0.673 | 15.45 | 0.049 | 7 | 71% |
| 30 | 0.772 | 0.770 | 0.708 | 0.769 | 0.816 | 0.805 | 0.773 | 17.75 | 0.038 | 5 | 81% |
| 40 | 0.998 | 0.847 | 0.784 | 0.814 | 0.868 | 0.844 | 0.859 | 19.73 | 0.074 | 9 | 91% |
| 60 | 0.861 | 0.877 | 0.827 | 0.880 | 0.952 | 0.891 | 0.881 | 20.24 | 0.041 | 5 | 94% |

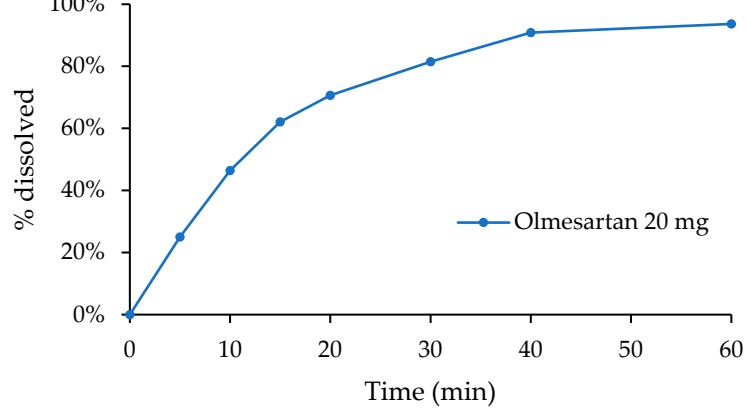

## 4. Conclusions

Olmesartan medoxomil tablets prepared by the direct compression method demonstrated an excellent dissolution profile related to the good solubility of the active substance which could be attributed to the selected excipients. The spectrophotometric analytical method developed demonstrated from the correct mixing during the manufacturing process to the concentration of olmesartan in the tablets.

SEM, DSC, and FT-IR studies as well as SeDeM galenical methodology demonstrated the proper selection of the excipients in terms of physical and chemical compatibility and suitability of the mixture for use in direct compression.

Finally, the critical quality attributes: physical characteristics, dimensions, content uniformity, breaking strength, friability, disintegration time and dissolution rate, demonstrated the suitable production process.

**Author Contributions:** Conceptualization, G.T. and M.Á.P.; methodology, R.G.; formal analysis, R.G.; investigation, G.T., M.Á.P. and R.G.; writing—original draft preparation, R.G.; writing—review and editing, G.T. and M.Á.P.; supervision, G.T. and M.Á.P.; funding acquisition, G.T. All authors have read and agreed to the published version of the manuscript.

**Funding:** This research received no external funding.

**Data Availability Statement:** Not applicable.

**Conflicts of Interest:** The authors declare no conflict of interest.

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
