# Peer review of "Formulation and Evaluation of Olmesartan Medoxomil Tablets"

_compounds, doi:10.3390/compounds2040028_

Round 1
Reviewer 1 Report
This is a detailed paper on formulation work based on olmesartan medoxomil tablets reported by R. Gonzalez. An interesting work. As with most papers, several concerns should be addressed as below:
1. Is lactose monohydrate directly used from vendor? The DSC does not conform to Form alpha or beta.
2. Have you ever considered using HPLC to measure the concentration in dissolution rate study? Since you have olmesartan medoxomil reference compound, it would be suggested to use HPLC to measure the concentration with low error.
Author Response
This is a detailed paper on formulation work based on olmesartan medoxomil tablets reported by R. Gonzalez. An interesting work. As with most papers, several concerns should be addressed as below:
- Is lactose monohydrate directly used from vendor? The DSC does not conform to Form alpha or beta.
The lactose monohydrate used is the provided by the supplier (Guinama S.L.U.). The DSC results show that the lactose used corresponds to alpha lactose monohydrate, according to the first three thermal events, with small traces of beta lactose anhydrous, corresponding to the last thermal peak.
- Have you ever considered using HPLC to measure the concentration in dissolution rate study? Since you have olmesartan medoxomil reference compound, it would be suggested to use HPLC to measure the concentration with low error.
This is a very interesting question. However, we consider the analytical method used to be highly sensitive with very low limits of detection and quantification, accurate with recovery rates above 95 %, and precise with a CV average of 1 %. Consequently, when performing the dissolution profile of tablets, standard deviations below 0.08 were obtained.
Reviewer 2 Report
It is necessary to proceed to make some corrections and expand some aspects.
• Page 9, line 308, it says 0.25%-0.5%. You are wrong: you should put 0.25%-5.0%.
• Table 9. It’s written in Spanish. It must be translated into English.
• Table 9. It seems to be the mean. It would be convenient to show the dispersion of the results or provide a new table with the individual results of the 6 tablets analyzed together with the mean, standard deviation, and CV%.
Author Response
It is necessary to proceed to make some corrections and expand some aspects.
- Page 9, line 308, it says 0.25%-0.5%. You are wrong: you should put 0.25%-5.0%.
The changes have been made.
- Table 9. It’s written in Spanish. It must be translated into English.
The changes have been made.
- Table 9. It seems to be the mean. It would be convenient to show the dispersion of the results or provide a new table with the individual results of the 6 tablets analyzed together with the mean, standard deviation, and CV%.
